# Cerebrovascular Events after Transcatheter Aortic Valve Replacement: The Difficulty in Predicting the Unpredictable

**DOI:** 10.3390/jcm11133902

**Published:** 2022-07-04

**Authors:** Oliver Maier, Georg Bosbach, Kerstin Piayda, Shazia Afzal, Amin Polzin, Ralf Westenfeld, Christian Jung, Malte Kelm, Tobias Zeus, Verena Veulemans

**Affiliations:** 1Department of Cardiology, Pulmonology and Vascular Medicine, Medical Faculty, Heinrich Heine University, 40225 Duesseldorf, Germany; oliver.maier@med.uni-duesseldorf.de (O.M.); georg.bosbach@uni-duesseldorf.de (G.B.); shazia.afzal@med.uni-duesseldorf.de (S.A.); amin.polzin@med.uni-duesseldorf.de (A.P.); ralf.westenfeld@med.uni-duesseldorf.de (R.W.); christian.jung@med.uni-duesseldorf.de (C.J.); malte.kelm@med.uni-duesseldorf.de (M.K.); zeus@med.uni-duesseldorf.de (T.Z.); 2CardioVascular Center (CVC) Frankfurt, 60389 Frankfurt, Germany; kerstinpiayda@gmail.com; 3Medical Faculty, CARID (Cardiovascular Research Institute Duesseldorf), Heinrich Heine University, 40225 Duesseldorf, Germany

**Keywords:** aortic stenosis, TAVR, percutaneous valve therapy, stroke, prediction, risk score

## Abstract

Background: Cerebrovascular events (CVE) are feared complications following transcatheter aortic valve replacement (TAVR). We aimed to develop a new risk model for CVE prediction with the application of multimodal imaging. Methods: From May 2011 to August 2019, a total of 2015 patients underwent TAVR at our institution. The study cohort was subdivided into a derivation cohort (*n* = 1365) and a validation cohort (*n* = 650) for risk model development. Results: Of 2015 patients, 72 (3.6%) developed TAVR-related CVE. Pre-procedural factors of our risk model were history of prior CVE, a larger aortic valve area (≥0.55 cm^2^), a large aortic angulation (≥48.5°), and enhanced calcification of the right coronary cusp (≥447.2 AU), left ventricular outflow tract (≥262.4 AU), and ascending thoracic aorta (≥116.4 AU). Our risk model was superior for in-hospital CVE prediction following TAVR in the establishment cohort (AUC 0.73, 95% CI 0.66–0.80; *p* < 0.001) compared to other risk scores, such as the EuroSCORE II or the CHA_2_DS_2_-VASc score. Conclusions: Although CVE prediction in patients undergoing TAVR is challenging due to the complex nature of the TAVR procedure, our study highlights that multimodal imaging is a promising approach to generate a more accurate risk model for CVE prediction.

## 1. Introduction

In the last decade, transcatheter aortic valve replacement (TAVR) has become the preferred alternative to surgical aortic valve replacement (SAVR) in patients with severe, symptomatic aortic valve stenosis (AS) who are intermediate- or high-risk candidates for surgery [1]. Technical advances of newer-generation prostheses and the increasing experience of operators have led to a progressive decrease in periprocedural complications and death following TAVR. Nevertheless, cerebrovascular events (CVEs) are still feared adverse incidents with devastating consequences for patients’ daily living and increased mortality [2,3,4]. The incidence of CVEs following TAVR is known to be related to patient-specific, procedure-related, and post-procedural factors [5], as summarized in Figure 1. However, to the best of our knowledge, there is no established risk model for prediction so far. Therefore, this study aimed to identify predictors of CVE, and to develop a risk model for in-hospital CVE following TAVR. The focus was on valvular and aortic calcium burden assessed by pre-procedural multimodal imaging as well as procedure-related factors.

## 2. Materials and Methods

### 2.1. Study Population

From May 2011 to August 2019, a total of 2015 patients underwent TAVR with transfemoral (*n* = 1694, 84.1%) or transapical access (*n* = 321, 15.9%) and self-expandable (*n* = 1399, 69.4%) or balloon-expandable prosthesis (*n* = 616, 30.6%) at our institution. In this single-center, retrospective analysis, only patients with pre-procedural contrast-enhanced CT assessment and entirely calculated CHA_2_DS_2_-VASC and HAS-BLED scores, as well as logistic EuroSCORE I, EuroSCORE II, and STS-PROM scores, were enrolled. TAVR was performed according to current guidelines between 2011 and 2019, under local anesthesia for TF access and general anesthesia for TA access. TF TAVR was performed with different generations of either the self-expandable CoreValve System (Medtronic Inc., Minneapolis, MN, USA) or the balloon-expandable SAPIEN System (Edwards Lifesciences, Irvine, CA, USA); TA TAVR was only performed with the balloon-expandable SAPIEN System.

### 2.2. Study Endpoints

The primary study endpoint was defined as in-hospital CVE. According to the Valve Academic Research Consortium-2 (VARC-2) criteria, CVE is an acute episode of a focal or global neurological deficit caused by ischemic, hemorrhagic, or undetermined etiology, and confirmed by neurological specialists or neuroimaging (i.e., computed tomography or magnetic resonance imaging). CVE was further classified in transient ischemic attack (TIA), defined as neurological deficit less than 24 h in duration, CVE with persistence of symptoms for more than 24 h, and detection of new cerebral lesions by neuroimaging.

### 2.3. Risk Score Assessment and Validation

For risk score development, we subdivided our population into a derivation cohort with TAVR from May 2011 to January 2018, including 1365 patients, and a validation cohort with TAVR from January 2018 to August 2019, including 650 patients. In the derivation cohort (*n* = 1365), we identified 60 patients (4.4%) with in-hospital CVEs. After exclusion of incomplete datasets, and using 1:10 propensity score matching with the variables age, sex, body mass index, and access route, we matched 56 patients with CVEs and 521 patients without any new focal or global neurological deficits. A modified CONSORT flowchart gives an overview of the patient population, selection process, and data analysis (Appendix A). For validation of the developed risk models, we performed another 1:10 propensity score matching of patients in the validation cohort (*n* = 650). Finally, 12 patients with in-hospital CVEs were matched with 120 patients in the non-CVE group (Appendix A).

### 2.4. Statistical Analysis

Continuous data are described as the mean ± standard deviation (SD) for normal distribution, and comparisons were performed using unpaired Student’s *t*-test and the Wilcoxon rank-sum test. Categorical variables are presented as frequencies and percentages, and comparisons were made using the chi-squared test and Fisher’s exact test. All statistical tests were 2-tailed, and a value of *p* < 0.05 was considered statistically significant. A multivariate logistic regression analysis using a purposeful selection of covariates was performed to determine independent predictors of in-hospital CVE following TAVR, including predictors with *p* < 0.05 in univariate analysis and those reported to have a well-known impact on CVE following TAVR by consensus opinion and previously published literature [3,5,6]. Receiver operating characteristic (ROC) analysis and Youden’s index—the point at which the value of ”sensitivity + specificity − 1“ is maximal—were used to find the optimal cutoff values for dichotomization of parameters containing continuous data. Results were reported as odds ratios (OR) with associated 95% confidence intervals (CI) and *p*-values. ROC analysis and areas under the ROC curves (AUCs) were performed to compare the new risk scores with other common risk models. All statistical analyses were conducted using SPSS version 23.0 (IBM SPSS Inc., Chicago, IL, USA).

## 3. Results

### 3.1. Baseline Patient Characteristics

In the derivation cohort (*n* = 1365), the patients’ mean age was 82.2 ± 5.2 years, and 56.3% were female. Patients in the CVE and non-CVE groups did not differ concerning cardiovascular risk factors, pre-existing antiplatelet and anticoagulation medication, or perioperative risk profiles (EuroSCORE II, STS Score). The only differences were observed in the history of prior CVE (CVE 17.9% vs. non-CVE 8.6%; *p* = 0.026), bleeding risk according to HAS-BLED score (CVE 3.7 ± 0.9 vs. non-CVE 3.5 ± 0.9; *p* = 0.048), and CVE risk according to CHA_2_DS_2_-VASc score (CVE 5.2 ± 1.1 vs. non-CVE 4.8 ± 1.2; *p* = 0.009). Compared to the derivation cohort, patients in the validation cohort (n = 650) differed in age (79.4 vs. 82.2 years), gender (48.5% vs. 43.7% male), history of prior CVE (16.7% vs. 9.9%), and risk profile (EuroSCORE II: 8.2% vs. 5.4%, STS Score: 7.6% vs. 4.5%). All baseline characteristics for the derivation and validation cohorts are shown in Appendix A, respectively.

In comparison to the validation cohort, patients in the derivation cohort were older (82.2 ± 5.2 years vs. 79.4 ± 6.7; *p* < 0.001) and had significantly higher surgical risk (STS score 7.6 ± 6.9% vs. 4.5 ± 3.1%; *p* < 0.001) due to multimorbidity, with a higher proportion of chronic diseases (e.g., arterial hypertension, pulmonary hypertension, peripheral vascular disease, reduced LVEF) and, therefore, prolonged hospital stay (11.6 ± 8.1 days vs. 10.0 ± 7.4 days; *p* = 0.040). All comparative baseline characteristics between the derivation and validation cohorts are shown in Appendix A.

### 3.2. Study Endpoints

Of the 56 patients with CVEs in the derivation cohort, 15 (26.8%) developed TIA, and 41 (73.2%) suffered from CVEs with persisting neurological deficits and new cerebral lesions in neuroimaging. Of the 41 CVEs involved, 29 had ischemic, 2 hemorrhagic, and 10 unknown causes. Disabling CVE was defined as a modified Rankin Scale (mRS) ≥ 2 points, and occurred in 48.8% (*n* = 20). The in-hospital stay was prolonged in patients with CVEs compared to the non-CVE group (CVE 18.7 ± 14.1 days vs. non-CVE 10.9 ± 6.8 days; *p* < 0.001). In the validation cohort, the CVE group (*n* = 12) was equally subdivided into TIA (*n* = 6, 50.0%) and stroke (*n* = 6, 50.0%).

### 3.3. Model Development in the Derivation Cohort

The results of the univariate logistic regression for pre-procedural, intra-procedural and post-procedural parameters are presented in Table 1. Significant pre-procedural predictors of in-hospital CVE included prior history of CVE (OR = 2.30, 95% CI 1.09–4.86; *p* = 0.029) as well as high calcification of the ascending aorta (OR = 2.44, 95% CI 1.32–4.52; *p* = 0.004) and the LVOT (OR = 2.48, 95% CI 1.08–5.66; *p* = 0.032). Intra-procedural predictors included post-dilatation (OR = 2.26, 95% CI 1.19–4.30; *p* = 0.013) and snaring (OR = 6.60, 95% CI 1.81–24.15; *p* = 0.004), while aortic regurgitation above grade I (OR = 3.29, 95% CI 1.29–8.35; *p* = 0.012) and new pacemakers (OR = 2.98, 95% CI 1.04–8.5; *p* = 0.041) turned out to be post-procedural predictors of CVE following TAVR. Covariates protective against in-hospital CVE included high intima–media thickness (OR = 0.01, 95% CI 0.00–0.12; *p* < 0.001), application of protamine (OR = 0.20, 95% CI 0.08–0.46; *p* < 0.001), and medication with clopidogrel after TAVR (OR = 0.50, 95% CI 0.27–0.91; *p* = 0.023).

### 3.4. Risk Model I for In-Hospital CVE with Pre-Procedural Assessment

The results of the multivariate logistic regression for pre-procedural, intra-procedural, and post-procedural parameters are found in Table 2. For practical applicability before TAVR, we first created a manageable risk model with only pre-procedural determinants. Regarding significance in univariate analysis and the current literature, six parameters were identified for final analysis, namely, history of prior CVE (*p* = 0.008), aortic valve area (*p* = 0.783), aortic angulation (*p* = 0.005), and MSCT-derived calcification measurements of the RCC (*p* = 0.041), the left ventricular outflow tract (LVOT) (*p* = 0.067), and the ascending aorta (*p* = 0.702), scored using Agatston units (AU).

To simplify the risk model, metric risk model parameters were dichotomized by using ROC analysis and Youden’s index for the calculation of cutoff values. Finally, another multivariate logistic regression was performed with all dichotomized values, resulting in risk model I for pre-procedural assessment, with attribution of one point for each value (Table 3).

This resulted in a score with a minimum of zero and a maximum of six points. The risk score showed a good prediction of in-hospital CVE, with an area under the curve of 0.73 (95% CI 0.66–0.80; *p* < 0.001) (Figure 2A) and the best performance for scores of four and five points compared to the non-CVE group (*p* < 0.01) (Figure 3A). 

### 3.5. Risk Model II for In-Hospital CVE with Post-Procedural Assessment

To further evaluate the influence of intra-procedural and post-procedural factors on the development of in-hospital CVEs, we created expanded risk model II. Therefore, three intra-procedural parameters (non-use of protamine, aortic regurgitation above grade I, and snaring) and two post-procedural parameters (no clopidogrel and no anticoagulation following TAVR) were identified as independent predictors of in-hospital CVE (Table 4). This resulted in a larger score with a minimum of 0 and a maximum of 11 points. Risk model II showed an even better prediction of in-hospital CVE compared to risk model I, with an area under the curve of 0.79 (95% CI 0.73–0.86; *p* < 0.001) (Figure 2B), and best performance for scores of seven and eight points compared to the non-CVE group (*p* < 0.01) (Figure 3B).

### 3.6. Comparison to Other Risk Scores for CVE Prediction

For the derivation cohort, our new developed risk models I (AUC 0.73, 95% CI 0.66–0.80; *p* < 0.001) and II (AUC 0.79, 95% CI 0.73–0.86; *p* < 0.001) for pre-procedural and post-procedural assessment appeared to have better predictive values for in-hospital CVE after TAVR than previously used risk scores such as EuroSCORE II (AUC 0.50; 95% CI 0.43–0.58; *p* = 0.950), STS score (AUC 0.57, 95% CI 0.49–0.65; *p* = 0.12), CHA_2_DS_2_-VASc score (AUC 0.62, 95% CI 0.55–0.70; *p* = 0.004), or HAS-BLED score (AUC 0.59, 95% CI 0.51–0.69; *p* = 0.027) (Figure 2C, Table 5).

### 3.7. Validation of the New Risk Models 

The developed risk models I and II were applied to patients in the validation cohort (n = 132), aiming at the prediction of in-hospital CVE. While no difference in CVE prediction could be observed for risk model I with pre-procedural assessment (AUC 0.53, 95% CI 0.37–0.68; *p* = 0.77), risk model II with post-procedural assessment only narrowly missed a significant difference in CVE prediction (AUC = 0.60, 95% CI 0.43–0.76; *p* = 0.08). The results of the risk model I validation are shown in Figure 2D, while those of model II are shown in Figure 2E. In comparison to other risk scores for CVE prediction in the validation cohort, neither our newly developed risk models I and II nor any of the other four examined risk scores achieved statistical significance (Figure 2F, Table 6).

## 4. Discussion

The incidence of stroke and TIA was similar to that reported in prior studies [3,4,5,7,8], with 3.6% of our patients developing new neurological deficits within the first few days after TAVR. This study demonstrates that the risk of CVE following TAVR is associated with (A) calcification burden of the device landing zone, namely, the RCC as a surrogate parameter of the aortic valve, LVOT, and ascending aorta calcification burden; (B) higher mechanical forces during the implantation process, caused by larger aortic angulation, snaring, and residual aortic regurgitation; and (C) lack of post-procedural dual antithrombotic or anticoagulation treatment. Our developed risk model was superior to both established risk scores for outcome prediction (EuroSCORE II and STS score) mentioned in the current guidelines [1], as well as the risk scores examined for CVE prediction after TAVR in previous studies (CHA_2_DS_2_-VASc score and HAS-BLED score) [9,10]. Nevertheless, the reliability of CVE prediction is limited due to the infrequency of this event. To the best of our knowledge, this is the first risk model developed for in-hospital CVE following TAVR including multimodality imaging for quantification of calcification burden, highlighting the importance of pre-procedural MSCT analysis.

### 4.1. Impact of the Aortic Root’s Calcification Burden

The more calcified the aortic and valvular structures, the higher the risk of debris embolization through mechanical manipulation with wires, catheters, the catheter-loaded device, and the new valve itself. In our analysis, independent predictors of in-hospital CVE following TAVR were higher calcium volumes of the aortic valve (predominantly of the RCC), the LVOT, and the ascending aorta. This relationship between the calcification burden of the aortic root and the probability of CVE was partially described previously. Tada et al. and Vlastra et al. described a higher risk of CVE following TAVR in patients with a heavily calcified aortic valve, without examination of the ascending aorta or LVOT [3,11]. Our study is the first to consider all parts of the aortic root for calcium quantification, resulting in pre-procedural cutoff values for RCC, LVOT, and ascending aorta calcium volumes, summarized by risk score model I. Previous studies reported small aortic valve areas with high pre-TAVR aortic peak gradients associated with early CVE risk, assuming high calcium volumes of the aortic valve without quantification by MSCT [2,6,8,12]. In our opinion, this is an incorrect conclusion in some cases due to different underlying flow patterns of aortic stenosis. In (paradoxical) low-gradient stenosis, for example, aortic valve calcium volumes might be low, yet the valve orifice area is small [13]. In our results, we could not confirm the relationship between small aortic valve area and the risk of CVE; on the contrary, larger aortic valve areas turned out to be predictive of CVE following TAVR in risk model I. This finding might be classified as selection bias due to relevance in previous studies, but it could also illustrate that not the aortic valve area but, rather, the calcium volume is predictive of CVE, because the AVA is small anyway in severe aortic stenosis.

### 4.2. Procedure-Related Factors

Most early CVEs following TAVR are supposed to be procedure-related due to valve positioning and the implantation process itself, causing disruption of calcific atheromatous debris from the aortic arch and the native valve. The impact of tissue embolization during TAVR was confirmed by histopathological analysis of captured debris in the cerebral protection filters of almost every patient examined [14,15]. Transcranial Doppler studies showed that most high-intensity transient signals, as a surrogate of microembolization, occurred during valve positioning and implantation [16], suggesting that the mechanical interaction between the transcatheter valve and the calcified native aortic valve plays a major role in periprocedural cerebral emboli. Furthermore, the number of solid emboli was correlated with the aortic valve calcium score assessed by MSCT [17]. In the present study, procedural factors associated with in-hospital CVE included snaring of the prosthesis stent frame after valve prosthesis embolization, and increased mechanical force due to large aortic angulation, resulting in higher rates of significant aortic regurgitation after TAVR. Several studies support our finding that valve dislodgment and repositioning are predictive of early CVE following TAVR [2,18]. Large aortic angulation is known to be associated with moderate-to-severe aortic regurgitation [19]. Therefore, the CHOICE trial showed that significantly more balloon post-dilatations were required after self-expandable valve deployment to manage varying degrees of aortic regurgitation, aiming to reduce periprosthetic leakage of the under-expanded valve [4]. In our study, balloon post-dilatation showed an increased risk of in-hospital CVE in univariate analysis but failed to become an independent predictor in multivariate analysis. Nonetheless, previous studies demonstrated an increased risk of CVE occurring within the first 24 h after the TAVR procedure as a result of the use of balloon post-dilatation [2,5,6,8].

### 4.3. Patient-Related Factors

History of prior CVE is a well-known predictor and the most common risk factor in studies investigating CVEs following TAVR [2,3,4,8,12,18,20], so it was included in our analysis as well. However, no other factors associated with an increased atherosclerotic burden were found to be associated with CVE following TAVR in our analysis, such as chronic kidney disease, diabetes, coronary artery disease, peripheral vascular disease, or carotid stenosis. One possible reason for this observation is the timing of CVE after TAVR. Nombela-Franco et al. classified the incidence of CVE after TAVR as acute (≤24 h), subacute (1–30 days), or late (>30 days) [8]. Most frequent thromboembolic events occur within the first day after TAVR, with about half of them presenting immediately or within the first few hours after implantation, and may therefore be considered procedural [8,14,20], as described in our analysis of in-hospital CVEs. However, subacute and late CVEs were associated with patient-related predictors, including factors associated with an increased atherosclerotic burden—as described above—and new-onset atrial fibrillation [6,8,12,20].

### 4.4. Antithrombotic and Anticoagulation Treatment after TAVR

There is still uncertainty regarding the optimal antithrombotic regime after TAVR. Although the ARTE trial suggested that single antiplatelet therapy after TAVR may be as effective as dual antiplatelet therapy for prevention of ischemic events, with a lower risk of bleeding [21,22], our results show a strong correlation between non-use of dual antiplatelet therapy with clopidogrel and in-hospital CVE in risk score model II. The same effect could be observed regarding non-use of non-vitamin-K direct anticoagulation (NOAC), regardless of indication, although the recently published GALILEO trial reported a higher risk of death, bleeding, and thromboembolic complications associated with rivaroxaban (10 mg daily) after TAVR compared to antiplatelet therapy [23]. Our results have already been verified in previous studies that recognized a higher risk of early CVE in TAVR patients without dual antiplatelet therapy [2,6] or NOAC [7].

Regarding primary prevention of CVE, patients with an atrial-fibrillation-related stroke showed benefit from pre-stroke statin in previous studies, with significantly lower neurological deficit compared to those without pre-stroke statin therapy [24,25]. These interesting results should be further explored in atrial-fibrillation-related cerebrovascular events following TAVR due to paused anticoagulation during the TAVR procedure or new-onset atrial fibrillation after TAVR.

Administration of protamine during TAVR decreased the risk of in-hospital CVE in risk score model II. This result seems to be a paradox, because the use of protamine antagonizes the effect of heparin during TAVR, and should therefore lead to a higher risk of thromboembolic events. Evidence for protamine use during TAVR with regard to CVE is rare, but protamine changes blood viscosity—an effect that can result in embolic as well as hemorrhagic cerebral infarction. Al-Kassou et al. evaluated the safety of protamine administration during TAVR with regard to bleeding complications as well as stroke [26]. As expected, the incidence of bleeding complications was significantly higher in the non-protamine group, but stroke also tended to occur more often without administration of protamine (3.6% vs. 1.5%, *p* = 0.08). This result supports our finding of the CVE-protective effect of protamine, without any sufficient explanation.

### 4.5. Limitations

Several limitations should be considered when interpreting this study. The risk scores used for comparison to our developed risk models were not created for the application of CVE prediction after TAVR, but served as broad clinical comparison models with considerable predictive value. As this was a single-center, retrospective study, there may exist confounders that were not accounted for in our analysis. There might be temporal bias due to comparison of non-contemporary derivation and validation cohorts with significant differences in baseline characteristics, due to the higher surgical risk of TAVR patients in the recent past. The small number of in-hospital CVEs limits the power of our multivariate logistic regression model, and selection bias due to matching factors cannot be excluded. Therefore, this type of risk model should be regarded as a hypothesis-generating approach that needs to be further developed into a risk model with a more satisfactory predictive value in larger, randomized, controlled multicenter trials and large prospective registry studies without the temporal bias of derivation and validation cohorts. At a minimum, we tried to consider the rarity of CVEs by choosing a small matching ratio of 1:10, in contrast to former studies, which usually chose a matching ratio of 1:4 for the investigation of CVEs after TAVR.

## 5. Conclusions

Our new risk models appeared to be the best for in-hospital CVE prediction after TAVR compared to other clinical risk scores previously used in the literature. However, an accurate method for reliable CVE prediction seems to be impossible so far, although many clinical and procedural risk factors are known. The expansion of TAVR therapy in severe symptomatic aortic stenosis towards younger and lower-risk populations with longer life expectancy will force us to discover the mechanisms determining CVE after TAVR. Although the evolution of the delivery system, refined patient selection, and better intra-procedural pharmacological protection may have contributed to a decrease in CVE incidence, the morbidity and mortality associated with CVE after TAVR are still high. Even if thromboembolic cerebral events remain neurologically inapparent, the effects on cognitive impairment and long-term dementia are not well understood. Therefore, larger multicenter trials with expanded neuroimaging are needed to quantify cerebral lesions, instead of waiting for the stochastic clinical effects of CVEs. Those patients who might benefit the most from transcatheter cerebral embolic protection devices should be identified by risk prediction.

## Figures and Tables

**Figure 1 jcm-11-03902-f001:**
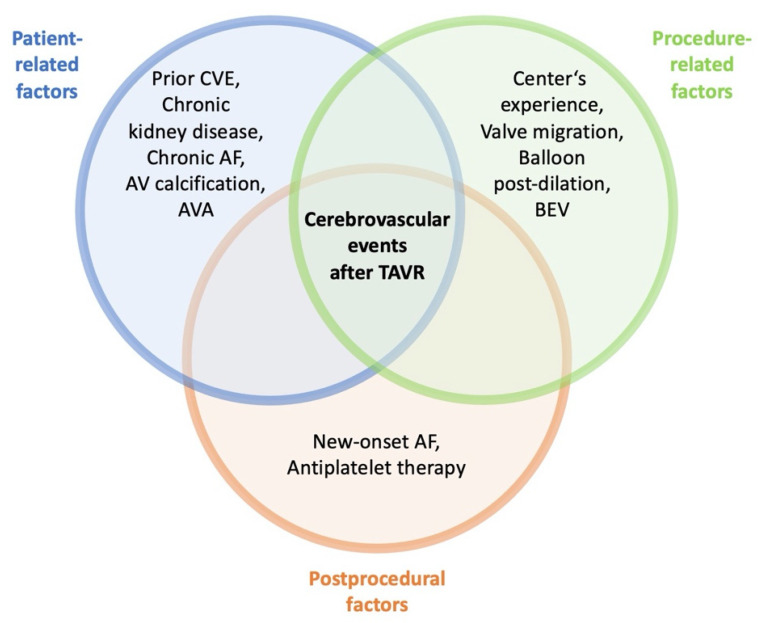
Previously identified factors associated with cerebrovascular events after TAVR with three overlapping categories: patient-related factors, procedure-related factors, and post-procedural factors. AF = atrial fibrillation; AV = aortic valve; AVA = aortic valve area; BEV = balloon-expandable valve; TAVR = transcatheter aortic valve replacement; CVE = cerebrovascular event.

**Figure 2 jcm-11-03902-f002:**
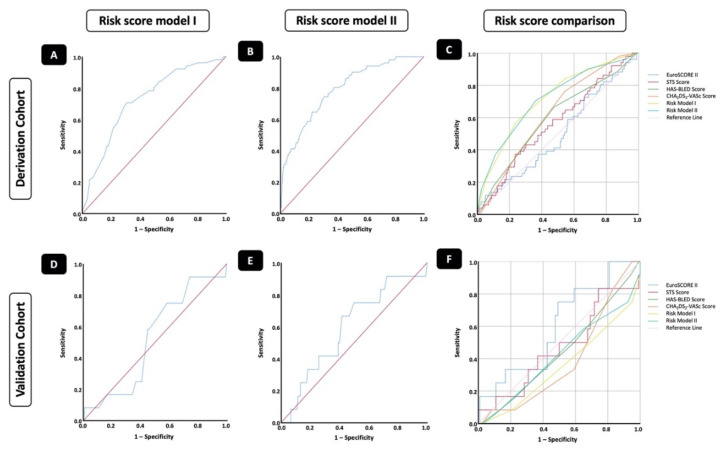
Risk score models I and II in comparison with established risk scores in the derivation and validation cohorts. (**A**) ROC analysis for risk score model I in the derivation cohort. AUC = 0.73 (95% CI 0.66–0.80), *p* < 0.001. Sensitivity = 70.6%; specificity = 69.0%; PPV = 19.5%; NPV = 95.7%. (**B**) ROC analysis for risk score model II in the derivation cohort. AUC = 0.79 (95% CI 0.73–0.86), *p* < 0.001. Sensitivity = 74.5%; specificity = 68.2%; PPV = 19.9%; NPV = 96.2%. (**C**) Comparative model discrimination for risk score models I and II, and established risk scores for the derivation cohort. (**D**) ROC analysis for risk score model I in the validation cohort. AUC = 0.53 (95% CI 0.37–0.68), *p* = 0.77. Sensitivity = 25.0%; specificity = 63.3%; PPV = 6.4%; NPV = 89.4%. (**E**) ROC analysis for risk score model II in the validation cohort. AUC = 0.60 (95% CI 0.43–0.76), *p* = 0.08. Sensitivity = 66.7%; specificity = 58.3%; PPV = 13.8%; NPV = 94.6%. (**F**) Comparative model discrimination for risk score models I and II, and established risk scores for the validation cohort. AUC = area under the curve; NPV = negative predictive value; PPV = positive predictive value; ROC = receiver operating characteristic.

**Figure 3 jcm-11-03902-f003:**
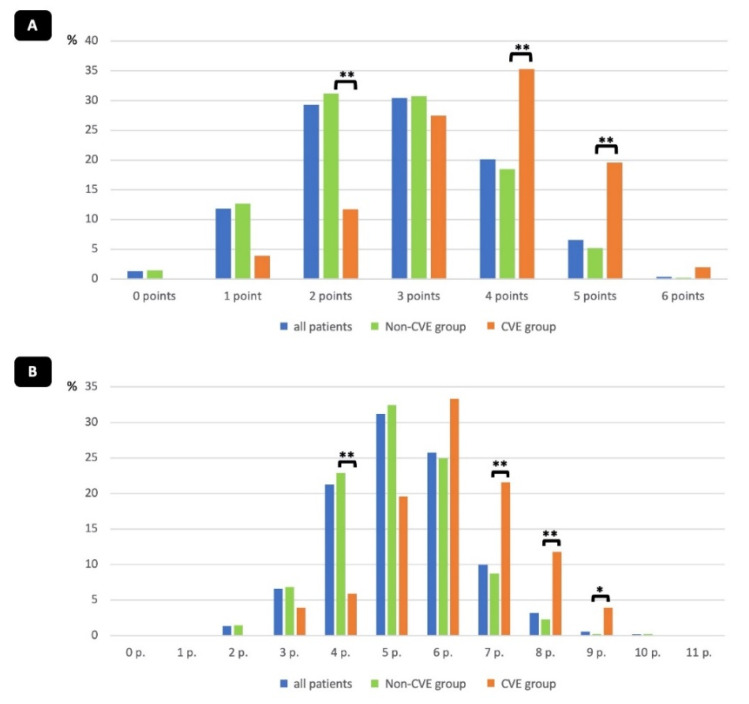
Percentage of patients in the derivation cohort stratified by risk score models I and II. Percentage of patients in each group stratified by risk score model I (**A**) and model II (**B**). Groups are shown with *p*-values (* *p* < 0.05, ** *p* < 0.01).

**Table 1 jcm-11-03902-t001:** Univariate logistic regression of pre-procedural, intra-procedural, and post-procedural parameters.

Parameter	OR	95% CI	*p*-Value
**Pre-Procedural**			
Atrial fibrillation	0.99	0.55–1.77	0.965
Porcelain aorta	0.87	0.20–3.79	0.849
Prior CVE	2.30	1.09–4.86	0.029 *
Prior dialysis	0.93	0.37–3.14	0.902
AVA (cm^2^)	0.94	0.44–1.98	0.866
Cardiac index (l/min/m^2^)	0.50	0.25–1.01	0.054
IMT (mm)	0.01	0.00–0.12	<0.001 ***
Annulus ellipticity index	0.93	0.85–1.01	0.088
LVOT area (mm^2^)	1.00	0.99–1.00	0.097
Aortic angulation (°)	1.03	1.00–1.06	0.072
AV Agatston score (AU)	1.00	1.00–1.00	0.089
RCC Agatston score (AU)	1.68	0.94–3.02	0.082
NCC Agatston score (AU)	1.00	1.00–1.00	0.048 *
LVOT Agatston score (AU)	2.48	1.08–5.66	0.032 *
Ascending aorta Agatston score (AU)	2.44	1.32–4.52	0.004 **
**Intra-Procedural**			
Prosthesis size (mm)	0.90	0.80–1.00	0.055
Self-expanding prosthesis	0.85	0.48–1.51	0.578
Procedure time (min)	1.00	1.00–1.01	0.361
Post-dilatation	2.26	1.19–4.30	0.013 *
Use of protamine	0.20	0.08–0.46	<0.001 ***
Valve dislodgement	1.10	0.38–3.23	0.860
Snaring	6.60	1.81–24.15	0.004 **
**Post-Procedural**			
Post-interventional AR ≥ II°	3.29	1.29–8.35	0.012 *
Clopidogrel after TAVR	0.50	0.27–0.91	0.023 *
(N)OAC after TAVR	0.54	0.28–1.05	0.068
Statin after TAVR	0.61	0.35–1.08	0.089
New pacemaker	2.98	1.04–8.50	0.041 *
*n* = 577 patients			

Parameters are shown with odds ratios (ORs), corresponding 95% confidence intervals (CI), and *p*-values (* *p* < 0.05, ** *p* < 0.01, *** *p* < 0.001). AR = aortic valve regurgitation; AU = Agatston unit; AV = aortic valve; AVA = aortic valve area; CVE = cerebrovascular event; IMT = intima–media thickness; LVOT = left ventricular outflow tract; NCC = non-coronary cusp; (N)OAC = (new) oral anticoagulation; RCC = right coronary cusp; TAVR = transcatheter aortic valve replacement.

**Table 2 jcm-11-03902-t002:** Multivariate logistic regression for pre-procedural, intra-procedural, and post-procedural parameters.

Parameter	OR	95% CI	*p*-Value
**Pre-Procedural**			
Atrial fibrillation	4.10	0.74–22.60	0.106
Porcelain aorta	5.62	0.40–78.83	0.200
Prior CVE	9.47	1.82–49.27	0.008 **
Prior dialysis	0.29	0.01–6.76	0.442
AVA (cm^2^)	1.03	0.82–1.30	0.783
Cardiac index (l/min/m^2^)	0.33	0.08–1.33	0.118
IMT (mm)	<0.01	<0.01–<0.01	<0.001 ***
Annulus ellipticity index	0.91	0.73–1.13	0.400
LVOT area (mm^2^)	0.99	0.98–1.00	0.004 **
Aortic angulation (°)	1.11	1.03–1.20	0.005 **
AV Agatston score (AU)	1.00	1.00–1.00	0.447
RCC Agatston score (AU)	5.76	1.08–30.83	0.041 *
NCC Agatston score (AU)	1.00	1.00–1.00	0.591
LVOT Agatston score (AU)	3.58	0.91–13.98	0.067
Central LVOT calcification	0.63	0.16–2.48	0.510
Ascending aorta Agatston score (AU)	0.79	0.24–2.64	0.702
**Intra-Procedural**			
Prosthesis size (mm)	1.32	0.90–1.93	0.158
Self-expanding prosthesis	0.15	0.02–0.91	0.039 *
Procedure time (min)	0.99	0.97–1.01	0.308
Post-dilatation	0.33	0.01–11.22	0.541
Use of protamine	0.03	0.00–0.24	0.001 **
Valve dislodgement	<0.01	<0.01–<0.01	0.996
Post-interventional AR ≥ II°	25.73	0.92–718.63	0.056
Snaring	105 × 10^9^	0.00–>105 × 10^9^	0.997
**Post-Procedural**			
Clopidogrel after TAVR	0.36	0.06–2.20	0.266
(N)OAC after TAVR	16.08	2.65–97.69	0.003 **
Statin after TAVR	5.32	1.18–23.99	0.030 *
New pacemaker	8.17	0.36–183.79	0.186
*n* = 345 patients, Nagelkerke R^2^ = 0.57, *p* < 0.001 ***			

Parameters are shown with odds ratios (ORs), corresponding 95% confidence intervals (CI), and *p*-values (* *p* < 0.05, ** *p* < 0.01, *** *p* < 0.001). AR = aortic valve regurgitation; AU = Agatston unit; AV = aortic valve; AVA = aortic valve area; CVE = cerebrovascular event; IMT = intima–media thickness; LVOT = left ventricular outflow tract; NCC = non-coronary cusp; (N)OAC = (new) oral anticoagulation; RCC = right coronary cusp; TAVR = transcatheter aortic valve replacement.

**Table 3 jcm-11-03902-t003:** Risk model I (pre-procedural parameters).

Parameter	OR	95% CI	*p*-Value
Prior CVE	1.94	0.85–4.43	0.114
AVA (≥0.55 cm^2^)	3.11	1.16–8.34	0.024 *
Aortic angulation (≥48.5°)	2.32	1.20–4.49	0.013 *
RCC Agatston score (≥447.2 AU)	1.80	0.94–3.44	0.077
LVOT Agatston score (≥262.4 AU)	2.01	1.08–3.75	0.028 *
Ascending aorta Agatston score (≥116.4 AU)	2.21	1.17–4.17	0.015 *
*n* = 532 patients, Nagelkerke R^2^ = 0.12, *p* < 0.001 ***			

Parameters are shown with odds ratios (ORs), corresponding 95% confidence intervals (CI), and *p*-values (* *p* < 0.05, *** *p* < 0.001). AU = Agatston unit; AVA = aortic valve area; CVE = cerebrovascular event; LVOT = left ventricular outflow tract; RCC = right coronary cusp.

**Table 4 jcm-11-03902-t004:** Risk model II (pre-procedural, intra-procedural, and post-procedural parameters).

Parameter	OR	95% CI	*p*-Value
Prior CVE	1.86	0.75–4.66	0.183
AVA (≥0.55 cm^2^)	3.18	1.11–9.13	0.031 *
Aortic angulation (≥48.5°)	2.49	1.24–5.01	0.010 *
RCC Agatston score (≥447.2 AU)	1.98	0.98–4.02	0.057
LVOT Agatston score (≥262.4 AU)	2.46	1.27–4.78	0.008 **
Ascending aorta Agatston score (≥116.4 AU)	2.28	1.15–4.49	0.018 *
Non-use of protamine	5.12	1.76–14.83	0.003 **
AR ≥ II°	2.77	0.91–8.42	0.072
Snaring	5.30	0.98–28.65	0.053
No clopidogrel after TAVR	2.64	1.22–5.72	0.013 *
No (N)OAC after TAVR	2.49	1.23–5.03	0.011 *
*n* = 532 patients, Nagelkerke R^2^ = 0.23, *p* = 0.006 **			

Parameters are shown with odds ratios (ORs), corresponding 95% confidence intervals (CI), and *p*-values (* *p* < 0.05, ** *p* < 0.01). AR = aortic valve regurgitation; AU = Agatston unit; AVA = aortic valve area; CVE = cerebrovascular event; LVOT = left ventricular outflow tract; (N)OAC = (new) oral anticoagulation; RCC = right coronary cusp; TAVR = transcatheter aortic valve replacement.

**Table 5 jcm-11-03902-t005:** Comparison of risk scores in CVE prediction after TAVR (derivation cohort).

Parameter	AUC	95% CI	*p*-Value
Modell I	0.73	0.66–0.80	<0.001 ***
Modell II	0.79	0.73–0.86	<0.001 ***
EuroSCORE II	0.50	0.43–0.58	0.950
STS score	0.57	0.49–0.65	0.120
HAS-BLED	0.59	0.51–0.69	0.027 *
CHA_2_DS_2_-VASc	0.62	0.55–0.70	0.004 **

AUCs, 95% CIs, and *p*-values (* *p* < 0.05, ** *p* < 0.01, *** *p* < 0.001) of the compared risk models. AUC = area under the curve; CI = confidence interval; STS = Society of Thoracic Surgeons.

**Table 6 jcm-11-03902-t006:** Comparison of risk scores in CVE prediction after TAVR (validation cohort).

Parameter	AUC	95% CI	*p*-Value
Modell I	0.35	0.18–0.52	0.092
Modell II	0.42	0.24–0.60	0.359
EuroSCORE II	0.60	0.44–0.76	0.251
STS score	0.47	0.29–0.65	0.716
HAS-BLED	0.44	0.28–0.61	0.514
CHA_2_DS_2_-VASc	0.39	0.24–0.54	0.204

AUCs, 95% CIs, and *p*-values of the compared risk models. AUC = area under the curve; CI = confidence interval; STS = Society of Thoracic Surgeons.

## Data Availability

The data presented in this study are available upon request from the corresponding author.

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
