# Peer review of "Cerebrovascular Events after Transcatheter Aortic Valve Replacement: The Difficulty in Predicting the Unpredictable"

_jcm, 2022, doi:10.3390/jcm11133902_

Round 1

Reviewer 1 Report

The authors aimed to develop a new risk model for CVE prediction with application of multimodal imaging. From May 2011 to August 2019, a total of 2015 patients underwent TAVR. The study cohort was subdivided into a derivation cohort (n = 1365) and a validation cohort (n = 650) for risk model development. 72 of 2015 patients (3.6%) developed TAVR-related CVE. Pre-procedural factors of authors risk model were history of prior CVE, a bigger aortic valve area (≥ 0.55 cm2), a large aortic angulation (≥ 48.5 °) as well as enhanced calcification of right coronary cusp (≥ 447.2 AU), left ventricular outflow tract (≥ 262.4 AU), and ascending thoracic aorta (≥ 116.4 AU). The authors risk model was superior for in-hospital CVE prediction following TAVR in the establishment cohort (AUC 0.73, 95% CI 0.66-0.80; p <0.001) compared to other risk scores like the EuroSCORE II or the CHA2DS2-VASc Score.

The article is well written and brings new knowledge to the discipline of neurocardiology.

Nevertheless, in the discussion section (4.4. Antithrombotic and anticoagulation treatment after TAVR), it is worth referring to studies that emphasize combined oral anticoagulant and statin therapy in the prevention of cardiocerebral incidents e.g. Choi K et al; Wańkowicz et al.

Author Response

We thank Reviewer #1 for his/her important comment that helped us to enhance the quality of our manuscript.

Point 1: In the discussion section (4.4. Antithrombotic and anticoagulation treatment after TAVR), it is worth referring to studies that emphasize combined oral anticoagulant and statin therapy in the prevention of cardiocerebral incidents e.g. Choi K et al; Wańkowicz et al.

Response 1: We acknowledge this point raised by the reviewer. Unfortunately, we missed emphasizing this evident part of medical stroke prevention by combined oral anticoagulation and statin therapy. We clarified these results in the discussion section according to the reviewer’s advice and added the suggested references (Choi et al. J Am Heart Assoc. 2019, Wańkowicz et al. J Clin Med. 2021). Therefore, patients with an atrial fibrillation-related stroke showed benefit from pre-stroke statin with significantly lower neurological deficit compared to those without statin therapy before stroke. These interesting results should be further explored in atrial fibrillation-related cerebrovascular events following TAVR due to paused anticoagulation during TAVR procedure or new-onset atrial fibrillation after TAVR. (please see 4.4. Antithrombotic and anticoagulation treatment after TAVR)

Reviewer 2 Report

Dear authors,

Thank you very much for the opportunity to review a paper with a high level of content and enrichment for our medical technical-scientific knowledge.

The wealth of information is predominant.

I would like to ask you to better describe the difference between the two types of Cohorts. It will make all the difference to whoever is reading the paper.

Second, was the ROC curve really satisfactory for the two risk models under analysis?

We have a number of selected patients that could be increased in order to give even greater significance to the new SCORE. A double-blind, controlled and randomized study would make all the difference and, above all, multicentric.

A meta-analysis on the subject could be raised.

I thank you for your attention and I am at your disposal.

Author Response

We thank Reviewer #2 for his/her important comments that helped us to shape further the message, hopefully improving the manuscript.

Point 1: I would like to ask you to better describe the difference between the two types of Cohorts. It will make all the difference to whoever is reading the paper.

Response 1: We thank the reviewer for this important suggestion. We have created a new Supplemental Table S3 in the supplementary materials section with a comparison of the patients’ baseline characteristics of the derivation cohort and the validation cohort, hopefully, more clearly defining the differences between both cohorts in this revised version of the manuscript. We modified the text in the results section to work out the differences in greater detail. Patients in the derivation cohort were older (82.2 ± 5.2 years vs. 79.4 ± 6.7; p<0.001) and had significantly higher surgical risk (STS score 7.6 ± 6.9% vs. 4.5 ± 3.1%; p<0.001) due to multimorbidity with a higher proportion of chronic diseases (arterial hypertension, pulmonary hypertension, peripheral vascular disease, reduced LVEF), and therefore, prolonged hospital stay (11.6 ± 8.1 days vs. 10.0 ± 7.4 days; p=0.040). We recognized these differences between both cohorts in the limitation section according to a temporal bias of two non-contemporary cohorts to develop and validate our risk score. (please see 3.1. Baseline patient characteristics, and 4.5. Limitations)

Point 2: Was the ROC curve really satisfactory for the two risk models under analysis?

Response 2: We agree with the reviewer that the ROC curve, especially for the validation cohort of risk model I (AUC 0.53) and risk model II (AUC 0.60), was not as satisfactory as it was for the derivation cohort (AUC 0.73 for model I and AUC 0.79 for model II). However, it was still more sufficient compared to all the other risk models (STS-Score, EuroSCORE, HASBLED, CHA2DS2VASC). As a result, we finally concluded that our risk model might be the best for CVE prediction after TAVR compared to other clinical risk scores used in literature before, but an accurate method for reliable CVE prediction seems impossible so far, although many clinical and procedural risk factors are known. If the ROC curve had been more satisfied with higher significance than other risk scores, our conclusion would have been more confident. However, in this case, our results should be regarded as a hypothesis-generating approach that has to be further developed into a risk model with a more satisfactory predictive value in larger multi-center trials without temporal bias of derivation and validation cohorts. We added this point in the limitations section. (please see 4.5. Limitations)

Point 3: We have a number of selected patients that could be increased in order to give even greater significance to the new SCORE. A double-blind, controlled and randomized study would make all the difference and, above all, multicentric.

Response 3: We thank the reviewer for raising this important limitation of this study. Our results have to be proven in further larger studies to validate the efficacy and to increase the significance of our risk score. For that matter, a randomized controlled multicenter trial would be the best way to realize that attempt. This point was added to the limitations section. (please see 4.5. Limitations)

Point 4: A meta-analysis on the subject could be raised.

Response 4: Thank you very much for your valuable suggestion. Indeed, during our literature search for cerebrovascular events following TAVR, we could not find any suitable meta-analysis about this topic after 2016 (Auffret et al., J Am Coll Cardiol 2016) regarding especially new-generation TAVR prosthesis. This is why we already started to summarize available studies and to create an updated meta-analysis about predictors of cerebrovascular events after TAVR with new-generation devices, hopefully, ready for publication within the next few months.

Reviewer 3 Report

Interesting report. It would be of help adding a multi- imaging panel depicting predictors of cardiovascular events.

  • The work provides interesting clinical insights. 
  • Appropriate population study and methodology.  
  • Results to be confirmed among patients enrolled in large prospective registry studies (add in the limitations section). 
  • It would be of help adding a multi- imaging panel depicting predictors of cardiovascular events. 

Author Response

We thank Reviewer #3 for his/her important comments that helped us to improve the manuscript substantially.

Point 1: It would be of help adding a multi- imaging panel depicting predictors of cardiovascular events.

Response 1: We created a graphical overview of previously identified factors associated with cerebrovascular events after TAVR in three overlapping categories: patient-related factors, procedure-related factors, and postprocedural factors. This new Figure 1 was included in the introduction section. We hope that this overview figure will be helpful for a better understanding of current knowledge in stroke prediction following TAVR. (please see 1. Introduction)

Point 2: Results to be confirmed among patients enrolled in large prospective registry studies (add in the limitations section).

Response 2: We thank the reviewer for raising this important limitation of our study. We added this point to the limitation section as an outlook for the need for further large prospective registry studies to investigate the important topic of stroke prediction following TAVR. (please see 4.5. Limitations)

Round 2

Reviewer 1 Report

The current version of the prepared manuscript meets the criteria for publication in the JCM